# How Will Digitalization Change Road Freight Transport? Scenarios Tested in Sweden

**Anna Pernestål** [1,*], **Albin Engholm** [1], **Marie Bemler** [2,3] **and Gyözö Gidofalvi** [1]

1   Integrated Transport Research Lab, KTH Royal Institute of Technology, 100 44 Stockholm, Sweden; aengholm@kth.se (A.E.); gyozo@kth.se (G.G.)
2   Scania CV AB, 151 48 Södertälje, Sweden; marie.bemler@scania.com
3   House of Innovation, Stockholm School of Economics, 113 83 Stockholm, Sweden
*   Correspondence: pernestal@kth.se; Tel.: +46-73-765-2443

**Abstract:** Road freight transport is a key function of modern societies. At the same time, road freight transport accounts for significant emissions. Digitalization, including automation, digitized information, and artificial intelligence, provide opportunities to improve efficiency, reduce costs, and increase service levels in road freight transport. Digitalization may also radically change the business ecosystem in the sector. In this paper, the question, "How will digitalization change the road freight transport landscape?" is addressed by developing four exploratory future scenarios, using Sweden as a case study. The results are based on input from 52 experts. For each of the four scenarios, the impacts on the road freight transport sector are investigated, and opportunities and barriers to achieving a sustainable transportation system in each of the scenarios are discussed. In all scenarios, an increase in vehicle kilometers traveled is predicted, and in three of the four scenarios, significant increases in recycling and urban freight flows are predicted. The scenario development process highlighted how there are important uncertainties in the development of the society that will be highly important for the development of the digitized freight transport landscape. One example is the sustainability paradigm, which was identified as a strategic uncertainty.

**Keywords:** freight transport; future scenarios; intuitive logic; logistics; digitalization

## 1. Introduction

Road freight transport is a key function of modern societies. At the same time, road freight transport accounts for significant emissions and contributes to congestion. For example, in the European Union, transportation emissions make up 23% of total $CO_2$ emissions, while road transportation accounts for over 70% [1]; heavy-duty vehicles represent 25% of $CO_2$ emissions from road transportation in Europe [2]. In Sweden, road freight transport is the dominant mode for domestic transportation, with almost 90% of the total volume of goods being carried by trucks [3]. Therefore, to achieve the United Nations sustainability goals, a sustainable and efficient road freight transport sector is key.

Digitalization, including automation, digitized information flows, and artificial intelligence (AI), provides many opportunities to improve efficiency, reduce costs, and increase service levels in the road freight transport sector. Digitalization also has the potential to radically change the business ecosystem in the sector. However, the road freight transport industry is fragmented and has a competition structure that is strongly oriented toward minimizing costs, a situation that might slow down the rate at which new technology is adopted.

Previous studies have identified the potential benefits and barriers of digitalization for freight transport, including impacts on sustainability, business models, and implementation (see, e.g., [4–8]). However, to fully understand the effects of digitalization, realize its potential, and avoid rebound effects, researchers have called for holistic analyses that explore the cumulative impacts of digitalization on the freight transport landscape [9,10].

This paper aims to fill this gap by asking how digitalization will change the road freight transport landscape, creating four future scenarios using the exploratory scenario method Intuitive Logics [11,12]. The study has 2040 as the target year and uses Sweden as a case study. The scenarios are based on input from more than 50 experts in the freight transport domain. The scenarios aim to describe complex future developments and are qualitative in nature. For each of the four scenarios, the impacts of digitalization on the road freight transport sector are investigated, including, e.g., the development of emissions, vehicle kilometers traveled (VKT), ton-kilometers. Furthermore, the opportunities and barriers to achieving a sustainable transportation system in each of the scenarios are investigated. The four scenarios represent plausible developments of the road freight transport sector and provide a platform for discussions, business development, and research on how digitalization can be used to achieve a sustainable road freight transport sector.

The remainder of the paper is structured as follows: Section 2 provides an overview of the previous literature on scenarios for freight transport and the effects of digitalization. Section 3 presents the method and process to derive the scenarios. Section 4 presents the results, including the four scenarios. The paper ends with a discussion of the results in Section 5 and the conclusions in Section 6.

## 2. Review of Literature

This section presents previous literature on the effects of digitalization on road freight transport and on scenario planning for handling these uncertainties in the sector. In the literature review, as well as in the whole paper, we consider digitalization in a broad sense, and include for example the use of digitized data, connected vehicles, and automated driving.

### 2.1. Effects of Digitalization on Road Freight Transport

Opportunities and barriers: Digitalization creates many changes relevant to the freight transport sector, including, e.g., the circular economy [13], e-commerce and changed consumer behavior [14], new business models [5], and automation [15–17]. Cooperative Intelligent Transport Systems (C-ITS) can improve traffic flow, reduce fuel costs, and increase efficiency in the transport system [18]. Digitalization not only enables the optimization of current value chains but also the reorganization of the entire value chain [8]. With digitalization and connectivity, multimodal transport can be optimized [19] and efficiency for haulers can be increased [20]. Most previous literature investigates the potential benefits of digitalization, but Molero et al. [6] studied barriers to implementation. They found that standardization and compatibility with existing information and computer technology (ICT) solutions were the main barriers to achieving the potential of ICT solutions in the freight transport industry.

Effects of automated driving: At lower levels of automated driving, truck platooning is expected to provide several benefits [21]. The first large-scale use case for driverless trucks on public roads is expected to be long haulage on highways between logistics hubs, with manually piloted trucks performing the first and last mile [15,22]. This would result in a new organization of road freight transport towards a structure similar to current multimodal freight transport networks [23]. Anticipated long-term impacts include cost savings for the trucking industry [24–26], which will increase road transport volumes [27]. Further, truck utilization is expected to increase [28–30], which could result in smaller truck fleets. The development of automated trucks is also expected to have significant impacts on freight transport actors' business models [31] and operations [32]. Moreover, there are reports focusing on labor market issues related to automated driving in freight transport [30,33,34]. Slowik et al. [35] and Paddeu et al. [36] list potential opportunities and drawbacks of automated driving for the transport industry.

Impacts on the role of the driver: In addition to automation, where the driver might be removed from the vehicles, in general, digitalization is expected to change the role of the driver. For example, drivers currently spend more than 5% of their workday on

administrative tasks, which can be reduced using digitized data and logistics-related documentation [20].

Impacts on business models: Digitalization is likely to "result in a radical shift in ways of business thinking" [5]. That is, digitalization and connectivity will change the interface between retailers and customers [14], which will have impacts on freight transport. Boon and van Wee [37] and Birtchnell et al. [38] discuss the impacts of 3D printing on transportation and logistics and draw different scenarios for how 3D printing may unfold as a new form of manufacturing. Digitalization may be used to obtain synergies between passenger and freight transport, but there is a need for new business models to release these synergies [39].

Impacts on sustainability: The World Economic Forum [40] estimates that digitalization has the potential to reduce emissions from logistics by 10–12% by 2025, primarily due to optimization of the logistics chain, based on crowdsourcing and shared warehouse agreements. However, these sustainability effects naturally depend on how digitalization is implemented. Bieser and Hilty [4] compare methods for evaluating the sustainability impacts of ICT, and Kayikci (2018) lists 23 sustainability criteria and evaluates how the digitalization of freight transport can impact these criteria. In a case study of six transport companies in Turkey, Kayikci found that their focus was primarily on using digitalization to improve economic sustainability.

### 2.2. Scenario Planning for Digitized Road Freight Transport

Previous literature also shows that there are many uncertainties in how digitalization is expected to materialize within the freight transport sector [10]. As a comparison, studies of the passenger transport sector show great uncertainties in the effects of digitalization on energy consumption, from −30% to +20% [41]. To fully understand the effects of digitalization on freight transport, there is a need for a holistic, system-level approach [9,10,42]. In this paper, we develop future scenarios with the aim of investigating how different trends might interplay and unfold.

Liimatainen et al. [43] developed future scenarios for the Finnish freight transport sector, focusing on $CO_2$ emissions as a function of Gross Domestic Product (GDP). Muratori et al. [44] explore scenarios for changes in emissions levels. Winkler and Mocanu [45] investigate three future scenarios for road transport in Germany to identify policies to improve the sustainability of the road transport sector, and Liu et al. [9] studied the impacts of road transport on emissions, climate, and health in the United States. However, these existing studies have not focused on the effects of digitalization or how digitalization is implemented.

To summarize, there is an extensive body of literature on how the use of digitalization can improve freight transport and logistics. Among those who have studied the effects of digitalization, most have focused on the effects of one aspect of digitalization rather than the effects of a combination of solutions [7] or have focused on the past or present to evaluate what has happened [5,19] rather than exploring the uncertain future.

In this study, the aim is to fill that gap and to explore the impact of digitalization in general on the freight transport landscape given an uncertain future. To do so, we have applied the holistic method of scenario planning.

### 3. Method

### 3.1. The Scenario Development Process

Scenario development can be based on different approaches and techniques [11]. This paper uses an exploratory scenario approach, in line with the underlying assumption that the development of digitalization in road freight transport is inherently uncertain and that there is not one agreed-on future. Instead, it is acknowledged that several plausible futures need to be explored in order to be prepared for future events. The exploratory scenarios outline various plausible trajectories of development but are not intended to

represent the most probable course of events (predictive scenarios) or to assess how a preferable scenario could be achieved (normative scenarios) [12,46].

This paper uses the Intuitive Logics (IL) technique. IL is a well-established scenario development technique for exploratory scenarios [11,12]. The outcome of an IL study is a set of mutually exclusive scenarios. At the core of IL is an analysis of trends believed to influence the system under study. First, trends are identified and mapped according to their presumed strength of influence. Then, the level of uncertainty in how the trends may unfold is judged. Trends that are assessed as having a strong influence and a low uncertainty constitute "certain developments". Certain developments represent the processes and forces shaping the object of study into a different future state than its current state. The certain developments on their own may result in drastic changes to the system.

From among the trends that are deemed to have a high degree of influence on the system and to be highly uncertain, the two most uncertain and important trends are selected to represent "strategic uncertainties". Crossing the strategic uncertainties creates a 2 × 2 matrix, with each quadrant representing one scenario [11]. A scenario is thus a representation of the future, given certain developments and one particular combination of the outcomes of the strategic uncertainties. The scenarios are then named, elaborated on, and analyzed, in this case with a focus on the development of freight transport.

The scenario development process in this study is shown in Figure 1. One important part of the process was the expert group workshops, which engaged 52 experts from 33 different organizations related to the Swedish freight transport sector, see Table 1. Most experts participated in multiple workshops, and all experts participated in the development of all four scenarios. Between the workshops, an analysis group consisting of the authors, one domain expert, and two future strategists analyzed and compiled the results generated at each workshop.

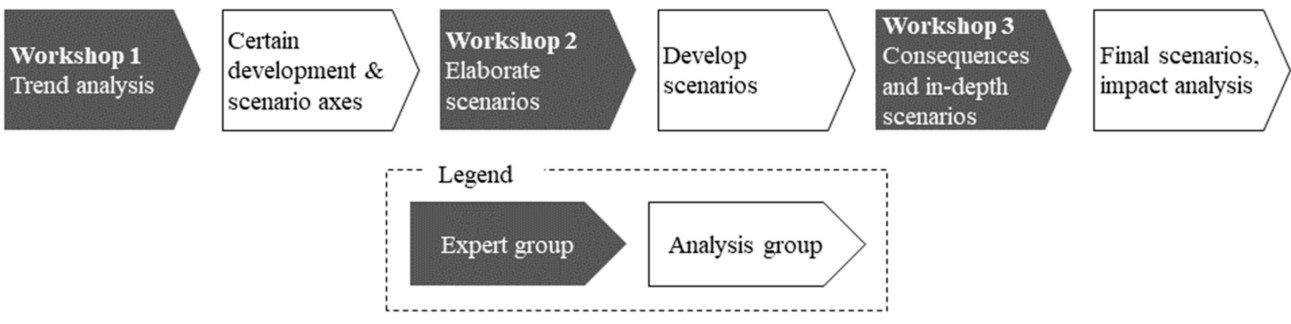

**Figure 1.** The scenario development process.

**Table 1.** The number of participants in the workshops from different types of organizations. One participant could join multiple workshops.

| Type of Organization | # Participants Workshop 1 | # Participants Workshop 2 | # Participants Workshop 3 | Total # of Unique Participants |
|---|---|---|---|---|
| vehicle manufacturer | 4 | 5 | 7 | 9 |
| logistic service provider or road carrier | 5 | 3 | 2 | 5 |
| municipality or region | 3 | 5 | 6 | 6 |
| academia | 3 | 3 | 4 | 6 |
| research institute or innovation network | 4 | 5 | 4 | 7 |
| national transport administration | 3 | 4 | 3 | 5 |
| transport buyer | 2 | 2 | 2 | 2 |
| digital service provider | 2 | 3 | 2 | 3 |
| real estate company | 2 | 1 | 1 | 3 |
| Consulting firm | 2 | 2 | 3 | 3 |
| industry association | 0 | 2 | 3 | 3 |
| Total | 30 | 35 | 37 | 52 |

The year 2040 was set as the target year for the study. However, this should not be seen as a precise year but rather a time horizon that is far enough in the future to be not affected by only incremental changes but not so far in the future that the whole of society can be expected to have changed.

### 3.2. Analysis of the Transportation Chain

For the analysis of the scenarios, a schematic overview of transport flows was used (see Figure 2), divided into the following six flows:

1. "Raw material" transport from extraction site(s) to product manufacturing site(s).
2. "Product" transport from manufacturing sites to retail locations (e.g., physical stores or e-commerce fulfillment centers).
3. "Transport to consumers" from retail locations. This transport may be handled by the end consumer (by driving a private car to and from a shopping center) or by an e-commerce distributor, through home delivery or delivery to a pick-up point.
4. "Material recycling" flow, where products are transported from the consumer (or recycling centers) to recycling plants.
5. "Product recycling" transport, where products are transported from the consumer for product refurbishment or to secondhand retail locations.
6. "Consumer to consumer (C2C) circulation" of products directly between consumers who sell and buy used products from each other or between consumers participating in services based on the sharing economy.

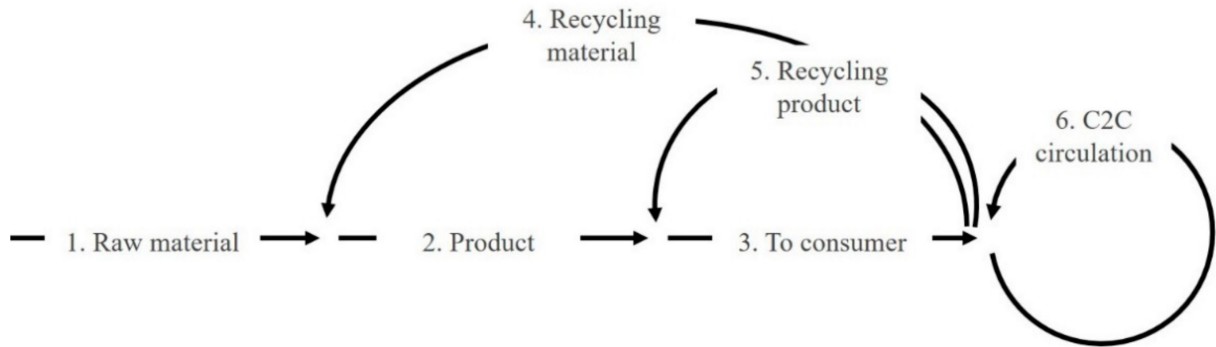

**Figure 2.** A schematic overview of the transport flows used to analyze the scenarios.

## 4. Results

### 4.1. Certain Developments

"Certain developments", based on the trends that were identified by the expert group to be highly probable to materialize, are presented in Figure 3. The certain developments are divided into trends external to the freight transport sector but affecting the transport sector (four areas), and trends within the freight transport sector (five areas). Brief descriptions of the nine areas are given in the following two subsections.

#### 4.1.1. Certain Developments External to the Freight Transport Sector

Within the area of urban planning, the main trends identified are urbanization and an increase in the size of regions. Urbanization is expected to lead to an increased focus on managing space limitations and efficient use of infrastructure, along with functions such as geofencing. (Geofencing is to create a virtual geographic boundary enabling software to trigger a response when a mobile device enters or leaves a particular area [47].)

Within data and technology, the development of AI, the Internet of Things, and automation are expected to play important roles. All new vehicles are expected to be connected, vehicles are expected to become increasingly automated, and—at least in specific

areas—driverless vehicles are anticipated. At the same time, the importance of data ownership and cybersecurity is expected to increase.

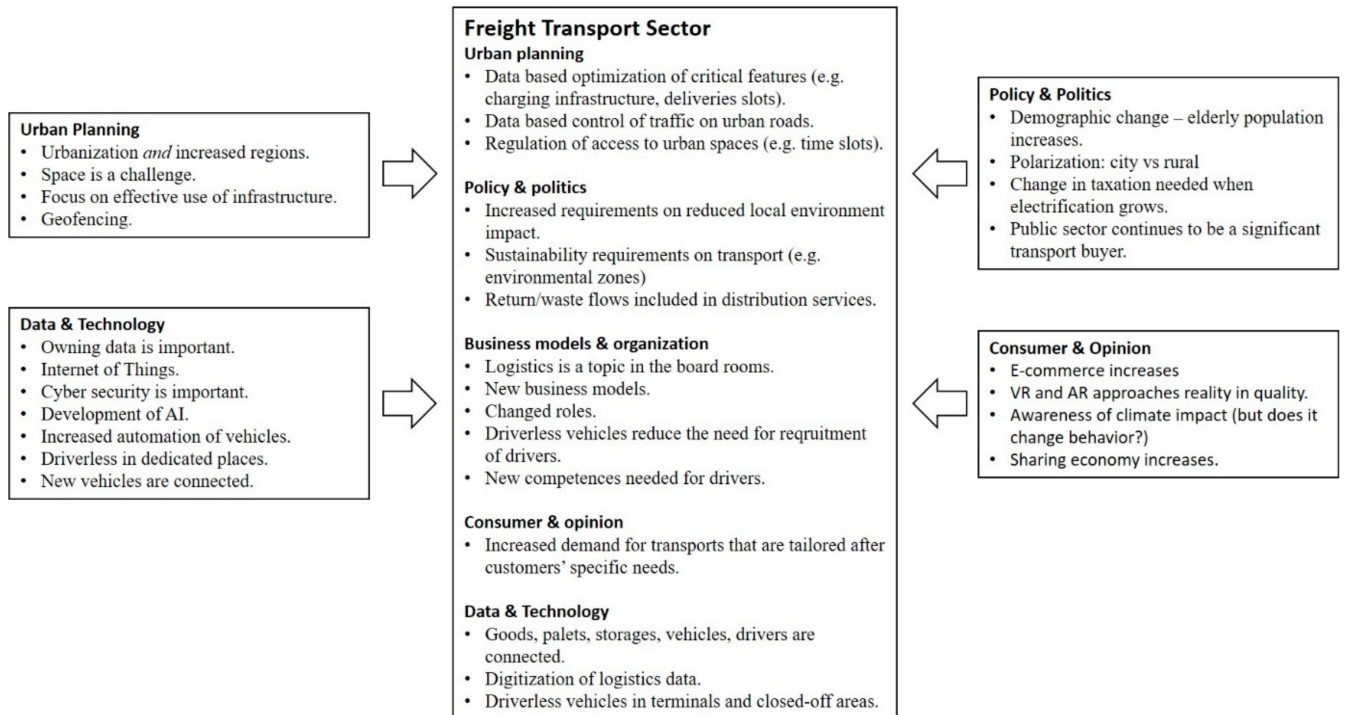

**Figure 3.** Certain developments, including external trends and trends within the freight transport sector.

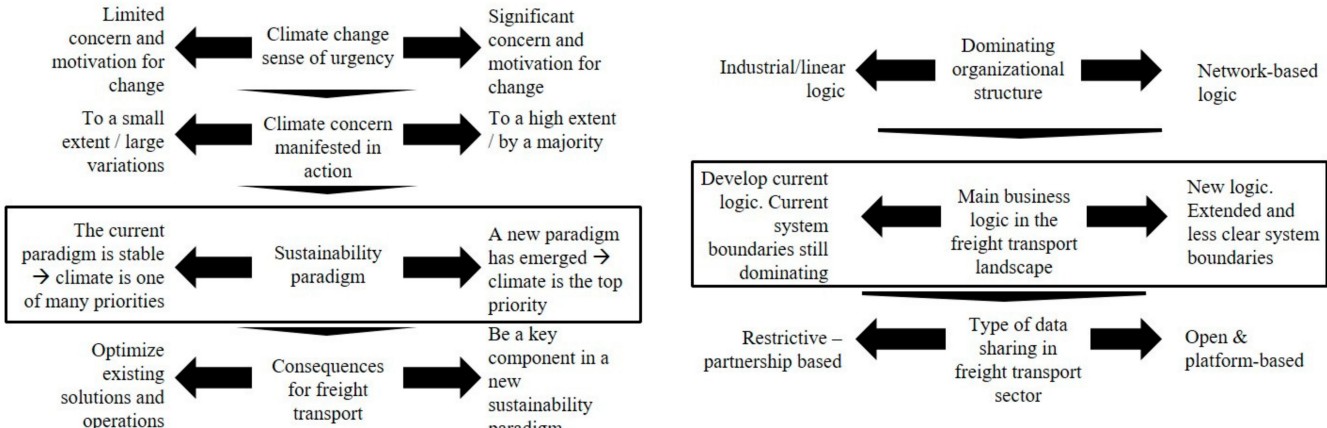

**Figure 4.** The strategic uncertainties used to form the scenarios.

Within the policy and politics area, demographic change, and an aging population are expected to be the most important trends. Urbanization, together with new types of services based on the sharing economy, are expected to blossom in densely populated areas and are predicted to create a polarization between cities and rural areas. A shift in taxation will be needed as fossil fuel vehicles are replaced by electric vehicles. The public sector is currently a significant transport customer and is anticipated to remain so. This means that the public sector will have the opportunity to put strict requirements on transport companies.

Within the consumer and opinion area, e-commerce, and the sharing economy are expected to continue to grow. Augments Reality (AR) and Virtual Reality (VR) will approach reality in terms of fidelity, which means that some physical experiences could be replaced

by virtual experiences. There will be a rising awareness of climate change, but expert groups are uncertain how (and if) this will affect peoples' behavior and opinions.

### 4.1.2. Certain Developments within the Freight Transport Sector

Urban planning trends that are expected to have a direct impact on the freight transport sector include the use of data to optimize the utilization of critical resources and reduce bottlenecks such as charging infrastructure and delivery slots. Data is also predicted to be used to control traffic flow on urban roads. There will be stricter regulation of vehicle access to urban spaces, which will be controlled using connectivity and geofencing.

Policy and politics will put increased requirements on transport to reduce local environmental impacts such as noise, particulate, and NOx emissions. This is expected to be implemented by using restrictions such as putting sustainability requirements on vehicles that transit through urban areas (called environmental zones [48]). Currently, distribution flows and return/waste flows are often handled by different organizations and following different business models. In the future, new business models that integrate these flows are expected. This would lead to increased utilization of vehicles.

New business and organization structures will develop, with new business models and changed roles for actors. Furthermore, logistics is expected to be seen as a strategic issue and to become a topic in the boardrooms of many actors. Automation and the development of driverless vehicles will affect businesses in terms of reduced costs for drivers, but there will also be a shift in the required skills for drivers and other personnel. Recruiting drivers is currently a major challenge for haulage companies, but automation is expected to reduce this need.

Within the consumer and opinion area, e-commerce and digitalization are predicted to lead to an increased demand for transport and deliveries that are both free of charge and tailored to customers' specific needs: e.g., delivery time and place.

Within the data and technology area, the digitization of logistics data, together with the increased connectivity and Internet of Things (IoT), will generate more data at various levels, including package, pallet, vehicle, and driver. This information will be used to design new services and improve effectiveness and service levels. It is uncertain how fast driverless vehicle systems will be developed and implemented, but it is considered certain that driverless vehicles will operate at least in terminals and on closed routes.

### 4.2. Strategic Uncertainties

Several uncertain trends were identified by the experts, and the analysis group found that these trends could be clustered around three themes: impact of climate-related actions, data sharing and business ecosystems, and the pace of technological development. The last of these three was selected to be included in the scenario descriptions, and the two first were selected as the two strategic uncertainties.

In the workshops, the two clusters representing strategic uncertainties contained trends at different levels of abstraction. The trends within each cluster were aligned by the analysis group, and the level of abstraction to be used to create the scenarios was selected. This level was selected to be a balance between changes in society in general and specific impacts on the freight transport sector. In Figure 4, the two themes for the strategic uncertainties are shown, and the levels selected to form the scenarios are framed.

One strategic uncertainty addresses the role and importance of climate issues in the sustainability paradigm. One outcome of this strategic uncertainty is the current sustainability paradigm, where climate, sustainability, economic aspects, and other societal issues (education, healthcare, security, etc.) are all more or less equally important and compete for attention and resources. The other outcome is that climate issues become the top sustainability priority, and climate awareness guides individual, business, and political decisions.

The other strategic uncertainty addresses the business logic in the sector. One outcome is that the current business logic is still present. Within this business logic, there is limited

willingness to share data outside the organization itself and its close partner network. In the other outcome, a new openness to sharing data changes the business ecosystem and enables a new network-based business logic.

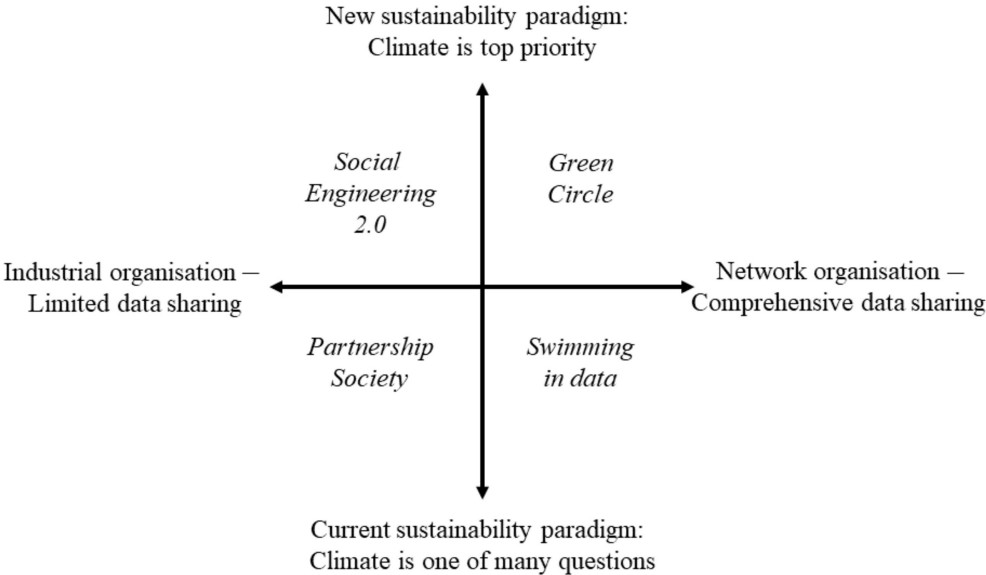

**Figure 5.** The four scenarios obtained by crossing the two strategic uncertainties.

*4.3. The Scenarios*

Combining the two strategic uncertainties gives four scenarios (see Figure 5). The four scenarios are labeled Social Engineering 2.0, Green Circle, Partnership Society, and Swimming in Data. In all four scenarios, the certain developments described in Section 3.1 constitute the background, but they are embodied in different ways depending on the outcome of the strategic uncertainties. The remainder of this section presents the four scenarios using fictional narratives, told from the perspective of 2040 looking back.

4.3.1. Social Engineering 2.0

Negative impacts from climate change during the 2020s created support for transformative policies. Since then, taxes on gasoline and diesel have multiplied, making fossil fuel vehicles so expensive that they are rarely used anymore. Today, in 2040, there are also heavy taxes on the extraction of non-biodegradable raw materials. Furthermore, a national digital "nudging" system has been introduced to incentivize more sustainable consumption. EU-level regulations forcing all manufacturing companies to track and recycle all materials used have been implemented.

Consumers perceive these developments with mixed feelings. There was and is strong awareness about the impacts of climate change, but willingness to decrease consumption was limited. This meant that producers addressed the climate issue by shifting to biomaterial-based production, and national and global demand for Swedish biomaterials skyrocketed. Moreover, product reuse and material recycling have grown drastically since the 2020s. Subscription and deposit models have made it easier to confirm consumption to current recycling regulations. Today, in order for companies to stay on top of logistics, they must keep track of their products after they are sold, as well as keep track of the components ordered for production.

Developments within AI and IoT have materialized in an abundance of sensors and data and have provided new decision support systems. Most people are aware of the business value of personal data. However, previous scandals in which personal data was shared with third parties for commercial and political purposes have created skepticism

about data sharing. Public entities have had to introduce data-sharing regulations that force companies to report emissions data.

In 2025, the Swedish parliament settled on a long-term transport agreement to promote climate-aware transport. Legislation on minimum fill rates on trucks was one of the interventions. It was first received with criticism, but over time this regulation has improved collaboration and efficiency through the use of connected goods. Another important policy was the decision to invest in electric roads for important freight corridors and European highways. Surging demand for biomaterial has increased the amount of road freight transport of timber and wood products.

The increase in recycling and product reuse has generated a new type of local transport flow. In the cities, a steady stream of used products is transported to new users, secondhand retail outlets, and recycling facilities. However, material recycling typically requires large-scale plants for economic reasons, and therefore, there are increased outflows from cities. Growing urban freight traffic, skepticism towards data sharing, and fill rate regulations make delivery times for e-commerce packages similar to delivery times in 2019. As a means to counter increased urban traffic, logistics is centered in "community hubs" that serve as both package pick-up points and drop-off points for waste and recyclables.

The bonus-malus policy incentivizing electric vehicles, together with governmental investments in electric infrastructure, has catalyzed the shift to an electric vehicle fleet. The new urban freight flows have led to a more diverse and flexible freight vehicle fleet. Automated vehicles are frequently used in the industrial flows of biomaterials to keep transport costs low.

The strong focus on data integrity has led to silo structures, where large actors from different sectors have created alliances. Data is shared only within these alliances. Transport actors are attractive partners due to their ability to gather data about goods and freight movement and conditions on roads and within cities.

### 4.3.2. Green Circle

Stopping climate change is not just one of many societal goals; it is the highest goal. Back in the early 2020s, powerful computer-based simulations and visualizations were able to clearly deliver the message showing the consequences of climate change. This sense of urgency empowered government, companies, and people in general to make tough decisions to meet the United Nations' global goals to stop climate change. Just as, in the EU, smoking indoors went from something taken for granted to something almost unthinkable during the 2010s, cities and streets without vehicles have been something people have become accustomed to and appreciative of. It is easy to buy sustainable products, thanks to taxes, subsidies, and government-supported labeling. The economic structure is now based on circular principles, and a radical reduction of material-based consumption has changed the way we produce, consume, and value our possessions. Selling something that you no longer use, mending something that is no longer working, and recycling things you do not need any more is easy, while throwing things away is expensive.

The circular economy has meant that the main type of transportation today is peer-to-peer, as local reuse and reselling has meant large flows of goods where people live. Previous long-distance goods transport that was almost invisible to consumers has shifted to short-distance logistics occurring in everyone's backyard. This greater visibility of transport has upset many citizens and has forced freight and logistics companies to come up with smarter and better ways of doing things. Data sharing has been the key success factor that has enable effective transport and use of the shared public space in cities. This was recognized by the government, which not only provided a platform for data sharing already in the early 2020s but also implemented laws that forced actors to share their data on this platform for the purpose of transparency of environmental impacts. The openness of data has provided opportunities to create new services and also provided tools for anyone to verify that services are sustainable.

Compared to the early 2020s, there has been a big change in focus within politics, with less attention to employment, equal rights, and welfare and more attention on the environment. This change has not gone by without protest, and it has been a turbulent time. Some citizens feel that they have sacrificed more than others, and large protests have become a part of the political landscape.

A wide diversity of actors is present, and the scene is dramatically different from how it looked at the beginning of the 2020s. New actors, as well as actors from other lines of business, are entering the logistic sector, where established actors are struggling to make money in the new data-intense era. Flexible and innovative small actors have changed how transportation is performed in both urban and rural areas.

### 4.3.3. Partnership Society

Back in the 2010s, start-ups in various fields tried to challenge big companies by providing new services. However, the intensive flow of news in the media highlighted issues with data security, and that data was abused to track people; this made people reluctant to share data. As a consequence, new services based on data did not take off. The large, already-established enterprises that could continue to build on their existing strong customer relations turned out to be the winners. These large companies realized the potential in utilizing customer data. They managed to change their businesses and engaged in strategic partnerships and alliances to increase their access to data and create new services. Today, in 2040, all producing companies (e.g., vehicles, furniture, etc.) also have significant business lines providing services based on insights from AI that uses the data collected from their product sales. In addition to these large companies, there are a number of large platform-based companies (including Airbnb, San Francisco, CA, USA and Logistics Cloud, Walldorf, Germany) providing links between suppliers and consumers.

For the past several years there have been significant and visible signs of a planet in ecological and social stress, but during the economic 2022 crisis the Paris Accord was forgotten, and the EU decided to prioritize actions to achieve a stable economy. In the years before 2020 there were signs of decreasing global trade, but that has shifted, and today global and regional trade is larger than ever. It is obvious that the climate situation is untenable, and the number of climate migrants is expected to be at an all-time high in 2040. This has led to new immigration challenges in many countries. To be politically viable, it is important that suggestions be beneficial for the economy, society, and welfare. Environmental friendliness alone is not a sufficient argument. The complex political landscape has led to short-sighted decisions by political parties, with the main focus on winning the next election and not political agreements with a long-term focus. All this has made progress on climate actions very slow, but one example is the shift to solar-powered electric vehicles, which turned out to be a very lucrative business.

Everyday life has not changed significantly since the late 2010s, but the new data-based services have made life much smoother. People and businesses are extremely aware of the risks of sharing data, but most consumers and companies are willing to take the risk to share their data if the service they get in return is of sufficient quality. Services include things such as custom-tailored clothes based on measurements provided by a "smart mirror" in the bedroom and proactive food deliveries based on the current contents of the smart fridge. There is a strong focus on how data is processed and shared with third parties, and the General Data Protection Regulation (GDPR) has been updated and is now even stricter than the first version, which came in 2018.

There are a number of parallel platforms owned by the main actors in the industry, providing effective logistics and deliveries. The largest actors during the 2010s are the ones who developed their own systems by adding data and AI. Flows are more effective than ever before, and both fill rate and route planning have reached new levels of efficiency. To survive in the business, partnerships are fundamental, both for larger companies, who can exchange data with their collaborators, and for smaller companies that need to

be allied with the main actors to get access to the platforms and the services provided on them.

Increased population and increased transportation demands have contributed to economic growth in the transport sector. Data-based solutions have made transport efficiency better than ever before, and service levels have increased significantly. Greenhouse gas emissions per ton-km have decreased, but due to the increase in transported goods, the volume of greenhouse gas and other emissions are still at the same level as they were back in 2020.

### 4.3.4. Swimming in Data

Economic inefficiencies of public organizations triggered an extensive public and political debate in Sweden during the 2020s. Inspired by examples of Dutch and Estonian municipalities that reduced their operating costs and created new and improved digital organizations and services, the government declared the concept of "network as a basis" for all public organizations and actions in 2026. This declaration empowered the Agency for Digital Government to carry out a reorganization of public authorities and increased the publication of data for authorities and commercial actors, creating proactive services in education and schools, healthcare, and crime prevention. The utility of data sharing has also manifested itself in several transport-related services: for example, geofencing has allowed the spread of deliveries and traffic across all hours of the day, and AI-powered predictive approaches for infrastructure maintenance have significantly reduced the number of unplanned service outages. Along with economic growth and efficiency, sustainability has also benefited from the network organization and comprehensive data sharing. Initiatives like "Sharing for the Earth," which was started in 2025, have triggered companies and organizations to share their data to achieve environmental and social sustainability goals.

Inspired by the new efficient and convenient solutions and services, citizens, organizations, and local governments are more open to sharing their data. To enable novel and genuinely effective and user-friendly solutions that utilize various kinds of data, several companies in Sweden and globally have joined the "The Digital Deal" agreement, which specifies how data collected from/about people, buildings, vehicles, utilities, and organizations should be shared and used. The few companies that did not submit to the data-sharing paradigm and did not dare to open up their data have lost their competitive edge as a consequence.

The volume of goods being transported is at an all-time high. E-commerce has grown dramatically. People expect deliveries to arrive just in time, tailored to their personal schedules and convenience. The last-mile delivery challenge has been resolved through a number of innovations, such as autonomous electric distribution vehicles, delivery robots, integrated pick-up/delivery boxes powered by digital locks, and modular multipurpose vehicles that can pick up recycling after delivering packages. In addition to enabling efficient consolidation of local distribution, data sharing and digitalization have become the keys to a number of AI-powered predictive solutions that combine user and consumer insights with information about transport flows and vehicle demand. For example, through predictive shipping and autonomous electric vehicles, the concept of the rolling warehouse has become a reality.

There are groups of citizens who are fed up with this fast-paced and largely convenience- and efficiency-driven society. Despite data sharing and sustainability initiatives, we have not managed to keep global temperature increases below the 2 degree Celsius goal. This has led to global political dissatisfaction, in a geopolitical landscape that has drastically changed due to the elimination of the fossil fuel-based economy.

In addition to the established transport actors that capitalized on the opportunities that are provided by the network organization and comprehensive data-sharing paradigm, digitalization also created new market entrants and partnerships. IT and social network giants not only became important via the customer insights that they can deliver but have also created the pay-as-you-go building blocks for cloud-based logistics that service

providers can build on. New global transport actors have appeared that—without having their own vehicle fleets, drivers, warehouses, or terminals—can create niche services with little investment. The opportunistic creation of such niche services is enabled by digital micro-contracts that allow simple and secure procurement of small, specialized transport assignments.

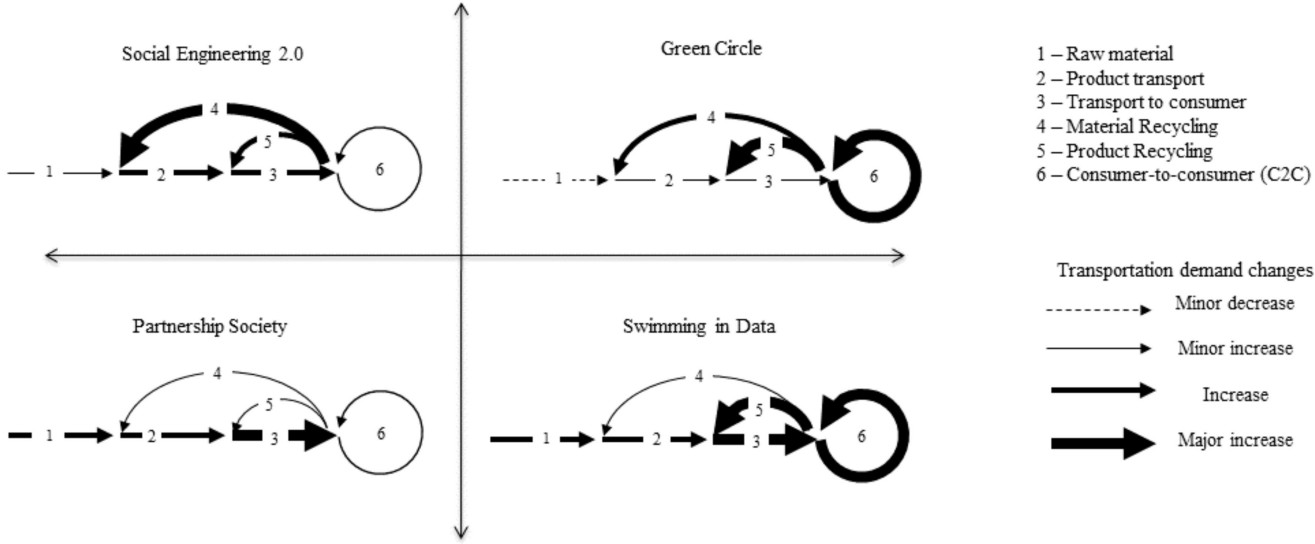

**Figure 6.** Expected changes in transportation demand in the four scenarios.

### 4.4. Transportation Flows in the Scenarios

This section presents the main characteristics of the transport flows in the different scenarios. The flows are visualized in Figure 6, and an overview is given in Table 2. These results are based on estimates by the expert group members, combined with the analysis group's analysis of the discussions during the export group workshops. Changes in volumes are in comparison to current levels.

#### 4.4.1. Transportation Flows in the Social Engineering 2.0 Scenario

Consumption is growing, but policies have shifted consumption towards products that are less energy- and material-intense to produce, as well as towards products made of biomaterials and recycled materials. Two factors explaining consumption growth are the growing population and the fact that people's consciences are less troubled when consuming products made of biomaterials—and thus consume more. Because of growing global demand for biomaterials, there is an increase in exports of forest-based biomaterials from Sweden.

Data sharing and collaboration within alliances of companies have made transport more efficient and increased consolidation levels and fill rates. These effects have primarily affected the industrial flows of raw material and products (flows 1 and 2 in Figure 6), where the demand is recurrent and only involves a limited number of actors. Recycling flows are much more difficult to optimize with the limited data sharing across actors that characterize Social Engineering 2.0.

Altogether, the Social Engineering 2.0 society shows increased transport demand in all transport flows, it is flows 2, 3, 4, and 5 that increase the most (see Figure 6).

**Table 2.** Effects of different parts of the transport chain in the four scenarios. The change in transport, in vehicle-kilometers, is marked as + or -, ranging from a major increase in demand (+++) to a major decrease in demand (—) compared to current levels.

| Flow | Social Engineering 2.0 | Green Circle | Partnership Society | Swimming in Data |
|---|---|---|---|---|
| 1<br>Raw materials | +<br>Increase in forest and other biomaterials for manufacturing Increasing export flows of biomaterials Shift to rail and sea to offset $CO_2$ emissions increase caused by volume growth | –<br>Increased reuse and recycling lead to decreases in raw material transport. | ++<br>Population growth and increased consumption increases demand Intermodal transport pushed by the government Automation and driverless vehicles in industrial flows. | ++<br>Transport demand increases, but data enables improved efficiency and fill rates Intermodality has increased due to data sharing |
| 2<br>Product | ++<br>Fewer products produced from new raw materials More products made of recycled materials are transported. | +<br>Fewer products produced from new raw materials Products are made from recycled materials. Data sharing enables high fill rates and effective transport. | ++<br>Population growth and increased consumption increases demand Limited data sharing limits efficiency improvements | ++<br>Increased demand due to e-commerce and population growth. Data sharing enables effective transport. |
| 3<br>To consumer | ++<br>Increase in consumption enabled by sustainable production and recycled materials | +<br>Increased distribution of recycled products Improved efficiency and fill rates due to data sharing | +++<br>Population growth and greater consumption increase demand.Limited data sharing limits efficiency improvements | +++<br>Increased demand due to more e-commerce and larger sharing economy Short delivery times and flexible deliveries increase transport demand |
| 4<br>Recycled Materials | +++<br>High taxes on raw materials encourages recycling | ++<br>Green policies foster increased recycling of material. | +<br>A limited increase, primarily driven by scarcity of raw materials. | +<br>A limited increase, primarily driven by scarcity of raw materials. |
| 5<br>Recycled Products | +<br>Limited product recycling | +++<br>Increased sharing and product recycling driven by sustainability policies and enabled by data sharing. | +<br>A limited increase, primarily driven by larger companies offering recycling of their own products. | +++<br>Companies offer many new services based on circular principles. Short delivery times and flexible deliveries increase transport demand |
| 6<br>Consumer to Consumer (C2C) | +<br>Policies make peer-to-peer sharing services attractive, but growth potential is constrained by limited data sharing | +++<br>New and growing flow between customers from resale/reuse of goods. New platforms and connections enable new possibilities for sharing and selling. | +<br>Growth potential is constrained by limited data sharing and lack of customer demand. | +++<br>Many new services for peer-to-peer sharing enabled by data sharing Short delivery times and flexible deliveries increase transport demand |
| Typical e-commerce delivery times (to consumer) | Several days | Hours—Days | Hours—Days | Minutes—hours |

### 4.4.2. Transportation Flows in the Green Circle Scenario

Changed consumer behavior, with increased demand for reused and recycled products, have significantly increased the demand for recycling transport (flows 4 and 5) and peer-to-peer transport (flow 6), but also for transport of products to consumers (flows 2 and 3). However, as a consequence of broad data sharing, in combination with strong sustainability regulations, fill rates and transport efficiency have improved. This is significant in flows 2 and 3, which are relatively predictable and where goods can be efficiently packed. Transport flows 4, 5, and 6 are much more difficult to improve on, both because they are less regular and because the goods being transported are typically bulky.

### 4.4.3. Transport Flows in the Partnership Society Scenario

The growth of the population and of e-commerce increases transport demand, but to attract customers, companies must deliver products and solutions that are convenient and tailored to customer needs. To be successful, strategic partnerships that include data sharing are crucial.

There is an increase in transport demand, primarily in flows 1, 2, and 3. For the transport of raw materials and products (flows 1 and 2), data can be shared within strategic partnerships to improve efficiency. This is more challenging in distribution (flow 3), as competitors operate in parallel instead of sharing data and space. In particular, there is increased competition over fast delivery times, which makes efficiency improvements even more difficult.

As there is limited interest in recycled products and no mandating legislation, these flows are limited. Further, limited data sharing constrains the potential of sharing services, leading to limitations in peer-to-peer flow as well.

### 4.4.4. Transport Flows in the Swimming in Data Scenario

In this scenario, the focus is primarily on using data to improve services and products for customers and consumers. This means short delivery times and extensive sharing services. Customer demand has increased, since the services are so convenient. This has led to increased transport demand in flows 1, 2, 3, 5, and 6. Improving efficiency is the second priority. The result is the efficient transport of materials and products (flows 1 and 2). For flows to and from consumers (3, 5, and 6), the importance of the customer experience—flexibility and short delivery times—limits the efficiency of the flows.

## 5. Discussion

As discussed in the introduction, the development of the road freight transport sector is one key to meeting sustainability goals. Therefore, in this section we discuss developments in the scenarios from a climate perspective.

### 5.1. Vehicle Kilometers Traveled

The expected development of vehicle kilometers is relevant to study, as it is related both to energy consumption (and thereby greenhouse gas emissions, in particular if vehicles use fossil fuels) and to congestion. The experts expect that digitalization may lead to increased consumption in general, as it enables new, customized services and products that are more attractive. At the same time, they expect digitalization to enable services based on the sharing and circular economies. These developments are expected to lead to increased vehicle kilometers traveled (VKT). Digitalization is, at the same time, expected to contribute to reductions in VKT by providing opportunities to set and monitor new regulations, such as kilometer-based taxes, and by increasing fill rates through data sharing and by connecting goods (Internet of Things). Digitalization may also improve route planning and optimization, as well as optimized the stowage of goods on vehicles. As shown in Figure 6, the outcome of these counterbalancing aspects depends on the scenario.

In the two scenarios on the left side of the scenario matrix, in which a traditional business logic predominates, flows 1–3 are central to development. These flows are stable,

predictable, and handled by relatively few actors—from one supplier to many delivery points—and thereby relatively simple to optimize.

In the scenarios on the right side, where a network-based business logic prevails, flows 4–6 are central to development. These flows are typically many-to-many, which makes them more challenging to optimize than flows 1–3. In addition, the goods handled are often bulkier because they are not efficiently packed (compare, e.g., transporting furniture from a factory efficiently packed to optimize transport and then assembled by the customer on arrival, versus used furniture that is already assembled and then transported between two users). On the other hand, flows 4–6 are often local or regional, and thus shorter than flows 1–3, and thus a shift from flows 1–3 to flows 4–6 might lead to a decrease in total VKT. To fully understand the impacts of digitalization on the total VKT, more detailed analysis and simulations are needed.

### 5.2. Climate Priority

As suggested in the scenarios, increasing the priority of the climate can unfold in two different directions. In the Social Engineering 2.0 scenario, the prioritization of the climate leads to increased use of bio-based raw materials such as forest products. In turn, this induces transportation demand in flows 1–3 in particular. Furthermore, it may also induce export flows if raw materials are exported to other countries.

In the Green Circle scenario, the prioritization of the climate leads to increased attention to the reuse and recycling of products, based on sharing and circular principles, and the focus shifts towards flows 4–6. As discussed above, this requires new optimization schemes and new challenges for transport related to bulkier goods. Another challenge with the shift towards flows 4–6 is that they are typically routed through areas where people live. Thus, transport will become more visible to citizens. One solution might be to use off-peak deliveries, where transportation takes place at night [49]. However, it should be noted that nighttime deliveries might have other sustainability impacts, such as noise pollution [49].

### 5.3. Limitations and Future Work

The work presented in this paper is based on the knowledge provided by the expert group during the scenario development process, and the process is qualitative and speculative in nature. The content in the scenarios and the transport demand estimations are, of course, dependent on the selection of experts, and certain trends or developments might have been overlooked. However, due to the relatively large number of experts and organizations represented in the work, and the unanimity among the experts, we believe that the results are representative and serve to provide a platform for dialog, business development, and research in order to achieve a sustainable future freight transport system.

To further explore the development in the scenarios, we propose that the development described in the scenarios be implemented in simulation models.

It should also be noted that the expert workshops were held before the COVID-19 pandemic, which changed travel and transport around the globe during 2020 and 2021. We believe that the major trends described in the scenarios have not changed significantly due to the pandemic but that certain trends, such as e-commerce, have increased their pace of development. Therefore, it would be interesting to revisit the scenarios after the COVID-19 pandemic.

## 6. Conclusions

In this paper, four exploratory scenarios describing how digitalization may affect the freight transport landscape have been developed using the Intuitive Logics method. During the process, two main uncertainties were identified: Whether a new sustainability paradigm with climate as the top priority will be present or not and whether traditional (hierarchical) business logic or network-based business logic will be dominant. By combining the different outcomes of the two strategic uncertainties, the four scenarios are formed.

The four scenarios should not be seen as the most probable futures but rather as plausible futures with "extreme" characteristics that provide a platform for discussion and development. The development of the transport sector is envisioned to be very different in the different scenarios. Three main takeaways are:

"Increased recycling flows": In three of the four scenarios, there will be an increased focus on recycling flows. In two of the scenarios, significant increases in recycling flows and peer-to-peer flows are predicted. This leads to new challenges in the optimization of flows and increases in freight transport in urban areas.

"Increase of VKT": In all scenarios, an overall increase in VKT is predicted, even in scenarios where the climate is prioritized. Handling this increase is important for achieving climate goals, including the development of renewable fuels, improved efficiency, and policies that target climate goals.

"External events and uncertainties": The scenario development process highlighted the fact that uncertainties in the development of society that are not directly related to either freight transport or digitalization are nevertheless quite important for the development of the digitized freight transport landscape. This is particularly clear in the strategic uncertainty related to whether the current sustainability paradigm will transform towards an increased focus on climate. Societal development will have a strong influence on what problems the digitalization of the freight transport sector should solve and what digital solutions are feasible.

**Author Contributions:** Conceptualization, A.P. and A.E.; formal analysis, A.P., A.E., M.B. and G.G.; funding acquisition, A.P.; investigation, A.P., A.E., M.B. and G.G.; methodology, A.P.; project administration, A.P.; writing of original draft, A.P. and A.E.; revision and editing, M.B. and G.G. All authors have read and agreed to the published version of the manuscript.

**Funding:** This research was funded by Strategic Innovation Program Drive Sweden, funded by VINNOVA, the Swedish Innovation Agency, the Swedish Research Council Formas, and the Swedish Energy Agency.

**Institutional Review Board Statement:** Ethical review and approval were waived for this study, due to the openness and non-personnel character of the study.

**Informed Consent Statement:** Patient consent was waived due to the openness and non-personnel character of the study.

**Data Availability Statement:** The data presented in this study are available on request from the corresponding author. The data are not publicly available due to privacy.

**Acknowledgments:** The authors would like to thank the experts that participated in the expert group; Jonas Eriksson, CLOSER, who participated in the analysis group; and future strategists Katarina Stetler and Erik Herngren, Kairos Future, who facilitated the workshops and the work in the analysis group.

**Conflicts of Interest:** The authors declare no conflict of interest.

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
