# Peer review of "How Will Digitalization Change Road Freight Transport? Scenarios Tested in Sweden"

_sustainability, doi:10.3390/su13010304_

Round 1
Reviewer 1 Report
The main contribution of this paper is its documentation of a systematic approach to generate structured input, from experts in fields related to transportation, regarding scenarios on the effect of digitization on road freight transport. The paper would be useful to readers interested in developing links between transportation academics and practitioners (a worthy and necessary endeavor) and those wishing to conduct research using insights from professionals in the field.
It is important to evaluate the paper on its own terms, i.e., with respect to what it sets out to do. It is qualitative and speculative (line 540 uses the term "explorative," which is appropriate -- "exploratory" would also work). The results (e.g., Section 4 and Table 1) should be seen specifically as a summary report of expert opinion generated by the process. I mention this because I was expecting, based on the abstract, a more quantitative paper, perhaps based on statistical modelling. I also found myself considering revisions to Section 4 and Table 1 as if I had participated in the workshops myself. A future reader of this paper might encounter similar issues.
To address this, I suggest highlighting the role of the workshops and experts more prominently in the abstract and in the introduction (beyond what is already in line 53). I also suggest that the terms "qualitative" and/or "speculative" be included in these locations, to emphasize the tentative nature of the workshop results, as is done in the conclusion with "explorative." I believe this would help to convey (what I understand to be) the intent of the paper.
Also, perhaps I'm missing this in the text, but did all of the experts participate in the development of all of the scenarios, or were they broken out into groups, each focusing on a specific part of the exercise? This would help the reader when comparing how the specific elements or discussion points relate across the scenarios (from top to bottom, and left to right of the 2x2 grid).
Finally, were there any interesting differences of opinions across the different types of experts, or between the expert group and the analysis group? Or was there unanimity? What was needed to reach consensus? This would help the reader to get a sense of how "firm" the results are. Would a different group of 50 experts reach different conclusions? It isn't necessary to add this discussion to the paper, but if there are any interesting observations, I encourage you to include them.
Overall, the paper reads well. I have a few specific comments:
-- Consider spelling out each abbreviation/acronym the first time it is used (AI, AR, VR, perhaps UN, etc.).
-- line 37, consider dominant instead of dominating.
-- line 39, consider deleting "the"
-- line 67, consider "By" rather than "With"
-- line 74, add "and" between "costs," and "increase"?
-- line 116, delete "be"
-- there are several places to address noun/verb agreement, singular/plural, has/have, etc.
-- line 253, "where" should be "were"
-- inconsistent identification of decades, e.g., 10s (line 328), 2020s (line 351), 2010's (line 356).
Reviewer 2 Report
Page 1, Title of the paper: “Will digitalization change the road freight transportation? - Future scenarios with Sweden as a case study”. My suggestion is to change the title to “Will (or “How”, please see my 2nd comment) digitalization change the road freight transportation? – Scenarios tested in Sweden” to shorten the long title.
Page 2:, lines 50-51: This paper aims to fill this gap by addressing the question “How might digitalization change the road freight transportation landscape?”. The title of the paper uses “Will” while the aim of the paper uses “How”. Please either use “how” in the title or “will” in the aim of the paper.
Page 1: “digitalization” appears twice in the key words.
Page 1, line 40: Please define all the acronyms/abbreviations within the text. For example, AI - Artificial Intelligence. Similarly, GDP, page 3, line 122, AR & VR, page 6, line 215, GDPR, page 10, line 385.
Page 2, lines 46-47: Please enhance the text which is associated to the specific 5 references [4-8] since “potential benefits and barriers” is a very interesting topic.
Page 2, line 64, “….and a conclusion in Section 6.”: Please change it to “….and the conclusions in Section 6.”
Page 2, Section 2. Review of Literature: My suggestion is to restructure the whole section and to present your findings under the following 3 subsections: digitized data, connected vehicles, automation (as the are defined at the beginning of Section 2).
Page 4, Figure 1: Please write the full word (instead of “dev.”).
Page 4, line 160: Please provide, within the text, some more details concerning the “expert group workshops” (e.g., when, who/organisations- perhaps in the form of a Table) apart from what is mentioned in the Acknowledgements.
Page 15, lines 537-538: “A solution might be to use off-peak deliveries where transports are performed during night time [49]”. Please also refer to the environmental impacts due to night deliveries (e.g., noise pollution).
Page 15: “Section 7. Conclusions” must be “Section 6. Conclusions”.
Page 15: “Section 7. Conclusions”: Please include a paragraph dedicated to the limitations and constraints of your research.
Finally, I would like to ask you why you did not consider the blockchain technology in your research.
